# Cold-Active Lipases and Esterases: A Review on Recombinant Overexpression and Other Essential Issues

**DOI:** 10.3390/ijms232315394

**Published:** 2022-12-06

**Authors:** Adamu Idris Matinja, Nor Hafizah Ahmad Kamarudin, Adam Thean Chor Leow, Siti Nurbaya Oslan, Mohd Shukuri Mohamad Ali

**Affiliations:** 1Enzyme and Microbial Technology Research Centre, Faculty of Biotechnology and Biomolecular Sciences, Universiti Putra Malaysia, Serdang 43400, Malaysia; 2Department of Biochemistry, Faculty of Science, Bauchi State University, Gadau 751105, Nigeria; 3Centre of Foundation Studies for Agricultural Science, Universiti Putra Malaysia, Serdang 43400, Malaysia; 4Enzyme Technology and X-ray Crystallography Laboratory, VacBio 5, Institute of Bioscience, Universiti Putra Malaysia, Serdang 43400, Malaysia; 5Department of Cell and Molecular Biology, Faculty of Biotechnology and Biomolecular Sciences, Universiti Putra Malaysia, Serdang 43400, Malaysia; 6Department of Biochemistry, Faculty of Biotechnology and Biomolecular Sciences, Universiti Putra Malaysia, Serdang 43400, Malaysia

**Keywords:** cold adaptation, 3D structure, esterase, lipase, psychrophilic enzymes, purification

## Abstract

Cold environments characterised by diverse temperatures close to or below the water freezing point dominate about 80% of the Earth’s biosphere. One of the survival strategies adopted by microorganisms living in cold environments is their expression of cold-active enzymes that enable them to perform an efficient metabolic flux at low temperatures necessary to thrive and reproduce under those constraints. Cold-active enzymes are ideal biocatalysts that can reduce the need for heating procedures and improve industrial processes’ quality, sustainability, and cost-effectiveness. Despite their wide applications, their industrial usage is still limited, and the major contributing factor is the lack of complete understanding of their structure and cold adaptation mechanisms. The current review looked at the recombinant overexpression, purification, and recent mechanism of cold adaptation, various approaches for purification, and three-dimensional (3D) crystal structure elucidation of cold-active lipases and esterase.

## 1. Introduction

Psychrophilic or extreme cold environments are usually characterised by diverse temperatures close to or below the water freezing point of 0 °C. The cold biosphere dominates about 80% of the Earth’s biosphere, and this environment could be a seasonal or permanent cold [1,2]. These cold habitats include fridges and freezers, high altitude alpine regions [1], permafrost areas [3], glaciers and deep seas [4], polar arctic and Antarctic regions [5]. The psychrophilic environment is inhabited by all the domains of life: Archaea, Bacteria, and Eukarya [6,7]. As cellular activities are disrupted at cold temperatures by high viscosity and low thermal energy [8], psychrophilic microorganisms require adaptive strategies to survive and thrive in such a harsh cold environment. Several adaptation strategies employed by psychrophilic organisms include inhibition of ice-recrystallization, nucleation of extracellular ice crystal by irreversibly binding to a particular plane of ice crystal, thereby preventing it from further secondary nucleation and continued ice growth, overcoming deficiencies in the uptake of carbon and nitrogen, and membrane fluidity maintenance [9]. One of the exciting survival strategies for cold-adapted environments that microbes use is the expression of cold-active enzymes that allow them to make an efficient metabolic flux at cold temperatures [3,10].

Cold-active enzymes are produced by psychrophilic microorganisms that are often heat-labile and perform a high catalytic activity at moderate to very low temperatures in contrast to their thermophilic and mesophilic orthologs [8,11]. The cold-active extremozymes generally achieved their efficient biochemical reactions by lowering both the enthalpy of activation and Gibbs free energy compared to their thermophilic and mesophilic counterparts [12]. Cold-active enzyme structures are homologous to their mesophilic counterparts. They only differ by discrete changes in their amino acid and spatial polypeptide structures, which are responsible for their distinct functions [3,13]. 

Cold-active enzymes have a high specific activity, low-affinity on the substrate at low temperatures, and they are structurally more flexible at their active sites; this flexible nature is due to weak intermolecular forces and increased exposure of hydrophobic residues [12,14,15]. Compared to their mesophilic and thermophilic counterparts, these features of high catalytic efficiency at low temperatures make these extremozymes highly attractive to the scientific community and provide potential applications in detergency, bioremediation, biofuels, and food industries [11,16,17]. Cold-active hydrolases such as protease, lipase, amylase, and cellulase were the most frequent enzymes characterised and used for industrial purposes compared to other cold-active enzymes [18,19,20,21]. Cold-active enzymes, in general, are ideal biocatalysts that can reduce the need for heating procedures, which improves the sustainability, cost-effectiveness, energy consumption, and quality of industrial production [11].

Esterase (carboxyl ester hydrolases, EC 3.1.1.1) and Lipases (triacylglycerol lipases, EC 3.1.1.3) are lipolytic enzymes that catalyse the synthesis and hydrolysis of acylglycerols, aryl, and carboxylic ester linkages [22]. Lipases and esterase are members of the serine hydrolase superfamily characterised with α/β hydrolase fold [23], made up of eight (8) β-strands with and six (6) α-helices that accommodate a highly conserved catalytic triad capable of nucleophilic reaction with their substrates [24,25]. Esterase catalyses the hydrolysis and synthesis of short-chain and partly soluble aliphatic esters. In contrast, lipases catalyse the hydrolysis and synthesis of long-chain fatty acid substrates that are water-insoluble [26]. These lipolytic enzymes were stable in organic solvents such as methanol, ethanol, DMSO, and n-hexane [27,28]. Furthermore, they showed high Regio- and stereo-preferences on diverse substrates, making them suitable biocatalysts for a wide range of industrial and commercial applications [29].

In recent years, there have been many articles published on cold-active enzymes [30,31,32]. Previous studies on cold-active lipases and esterases focused on their isolation from various sources, overexpression, and biochemical characterisation [33,34,35,36]. Still, their purification and three-dimensional (3D) structures have received little attention. This review focused on recent information recombinant overexpression, purification, 3D structure elucidation, and their mechanism of cold adaptation of cold-active esterase and lipases. Their isolation methods were also considered. Although only articles that reported on cold-active lipase and esterase were examined, most of the issues we came across were not specific to cold-active lipase and esterase but were valid to all enzymes. The summary of various sections of this review is depicted in Figure 1.

## 2. Approaches for Isolation of Genes Encoding Cold-Active Lipase and Esterase

The increasing industrial need for enzymes with high biochemical activity at low temperatures capable of synthesising biodiesel, biopolymers, chiral intermediates, and fine chemicals has led to the discovery of novel sources and screening methods for such extremozymes [16,37,38]. The cold-active esterase and lipase sources were reviewed [14,17,31,39]. The sources for these cold-active lipolytic enzymes were microbes of different life domains and species originating in diverse cold environments such as permafrost soil, polar regions, glaciers, and high-altitude mountain regions [40,41]. Although numerous cold-active esterases and lipases have been isolated and characterised, lipase from *Candida antarctica* B is currently employed in the industries [42,43,44]. Over the last few decades, the classic method of culture-dependency is used to isolate, screen, and discover cold-active lipases and esterases from the psychrophilic environment [45]. The standard way also involved isolation of microbes from natural surroundings, culturing or growing in the laboratory to determine the presence of the microbes, and gene sequencing to determine the order of nucleotides in DNA of the microbe.

The conventional culture-dependent approach is the backbone for many microbiological discoveries in academia and industry. The method is easy to handle and relatively cheaper. Examples in which this approach is applied include GBPI_Hb61 cold-active lipase [46], alkaliphilic cold-active esterase from arctic marine bacterium *Rhodococcus* sp [47], cold-active esterase (EstN7) from a *Bacillus cohnii* strain [48], cold-active lipase and esterase from Siberian permafrost *Psychrobacter* [49]. Furthermore, traditional laboratory methods of discovering and isolating novel lipases and esterases are time-consuming, and about 99% of microorganisms cannot be cultured through this conventional approach [46,50]. This setback is being addressed towards the application of metagenomics built upon the community or environmental genomics of uncultured microbes using experimental high-throughput sequencing technologies and bioinformatics tools that cover sequencing, metagenomic assembly, binning, domain prediction and pathway databases [40,51,52].

The metagenomic approach uncovered novel enzymes that often play a vital role in the biotechnology [53]. Metagenomics is often described as studying and analysing genomes found in natural habitats. Metagenomic libraries are DNA fragments extracted from environmental or community samples and cloned into specific vectors; the smaller fragments less than 15 Kb are constructed into plasmids, whereas larger inserts (25 to 200 Kb) are built in vectors such as cosmids and fosmids [54]. Functional metagenomics and sequence-based metagenomics provide information on enzymes’ evolutionary profiles, genomic linkages, and their functions [55]. Chauhan [56] and Dhanjal, Chopra [57] reviewed several novel enzymes discovered using a metagenomic approach, including lipases and esterases obtained from different environmental samples. Recently, a cold-active esterase PMGL3 was obtained from the metagenomic DNA library of Siberian permafrost, and other lipase genes have also been isolated and identified from various metagenomic libraries [36,58,59]. These studies showed that metagenomics is an effective technique for identifying, extracting, and discovering novel lipolytic enzymes. Despite the inspiring feature of the metagenomics technique in expanding our understanding of the evolution, ecology, diversity, and function of the microbial communities previously thought uncultivable, the method still encounters numerous challenges.

Dhanjal, Chopra [57] reviewed the significant challenges and solutions of the metagenomic approach; the issues of concern include the presence of fewer genes encoding enzymes of interest in metagenomic DNA, substrate scarcity for functional screening, low efficiency and enzyme performance in the artificial or induced approach, low screening efficiency of rare activities, a limited number of enzymes that perform efficiently in industrial, limited access to reliable bioinformatics tools to analyse large quantities of data sequence conditions robustly, and shorter reliable prediction tools for predicting enzyme activity on their coding sequence. Other issues include the bias associated with employing a heterologous host, usually *E. coli*, the host’s ability to express, fold, and produce the active enzyme [60]. These challenges need to be addressed to harness the power of these technologies and understand the biodiversity in our environmental samples. 

## 3. Cold-Active Lipase and Esterase Overexpression in Recombinant Heterologous Hosts

The most common strategy for obtaining large quantities of desired proteins is recombinant overexpression in a heterologous host [61,62]. Although the technique is often used in producing cold-active lipases and esterases, it is not specific to even cold-adapted enzymes but all recombinant proteins. When expressed in the cytosol, recombinant proteins are often produced at a greater yield, but they may also be regulated to be released into the culture media [61]. The overexpression of recombinant cold-active lipase and esterase is often achieved using mesophilic expression systems such as *E. coli* [63], yeast [64], and insects [65]. The production of large quantities of such enzymes at high concentrations remains challenging. As for other cold-active enzymes, the temperatures that cold-active lipase and esterase require for proper folding is inconsistent with the optimal growth temperature of these expression hosts [11]. The typical approach to mitigate folding problems in *E. coli* is to reduce the post-induction temperature below 20 °C. However, this slows down the host growth rate and the heterologous enzyme’s synthesis rate. Table 1 summarises some recently reported overexpression of cold-active lipase and esterase in a recombinant heterologous host. 

*E. coli* was selected as the preferred expression host, and just one of the enzymes was produced in *Saccharomyces cerevisiae* (*S. cerevisiae*). However, the *E. coli* Rosetta ^TM^ strain was reported to be used once [66], BL21 (DE3), was the most popular. Other Gram-negative bacteria, such as *Pseudomonas* and *Burkholderia*, lack suitable promoters and require foldase (a special chaperon) and extracellular fatty acids to induce their expression, a mechanism that is primarily unclear [67,68]. The two most common yeasts used for expression systems were *S. cerevisiae* (Baker’s yeast) and *Pichia pastoris* (*P. pastoris*). Its major drawback is its strong natural tendency of S. cerevisiae to ferment carbohydrates to ethanol, which is toxic at low culture density. However, *P. pastoris* lacks the problem of harmful ethanol synthesis, but it cannot express any gene of interest. While specific proteins may have no issues with being expressed, others may have problems associated with glycosylation, secretion, and folding [69,70]. A recent study on recombinant overexpression by Xue, Yao [64] found excellent expression of cold-active esterase in the *S. cerevisiae* heterologous host, which was attributed to similarities between the yeast family to which the wild gene and *S. cerevisiae* belongs. Since the carbon source was n-propanol and isobutanol and not sugars, the limitation of using *S. cerevisiae* was not mentioned. Another heterologous host for recombinant proteins is insect cell culture systems, which are well-known for their use in creating vaccines and viral insecticides [71,72]. Compared to other eukaryotic expression systems, high levels of heterologous gene expression are frequently achieved, especially for intracellular proteins [73]. In several instances, the recombinant proteins are soluble and easily collected from infected cells [73,74]. In one study, a *Yarrowia lipolytica* (LIPY8) extracellular lipase gene was expressed using a baculovirus expression system in insect cells, and it was interesting that the best pH and temperature for cold-active lipase LipY8p expressed in insect cells were very different from those for the same enzyme expressed in *P. pastoris* [65]. Moreover, it is too early to conclude how the change in heterologous host from yeast to insect increases the cold activeness of a particular enzyme. On the other hand, adaptability to a wide range of culture broths and its rapid growth and high enzyme yield were the major favourable characteristics that allowed the utilisation of *E. coli* for recombinant overexpression of heterologous proteins [75,76]. The major disadvantage of using *E. coli* host is the production of bodies [77].

Inclusion bodies are insoluble protein aggregates that lack biological function [78]; their formation often occurs when eukaryotic proteins are overexpressed in a heterologous host such as *E. coli* [79]. Inclusion bodies have been considered a significant obstacle to producing soluble and active recombinant proteins [80,81]. In Table 1, most of the cold-active lipase and esterase were overexpressed in soluble forms, and only five (5) were produced as insoluble or soluble but in inactive forms. It is difficult to explain why most articles we examined in this review reported more soluble expression than insoluble inclusion bodies. Furthermore, there has been a great success not only in using biochemical and molecular techniques to prevent their formation or to address various challenges during their isolation, solubilisation, refolding, and purification [80], but their biological activity is also emerging [82,83] contrary to the previous notion that they lack activity [78].

**Table 1 ijms-23-15394-t001:** Cold Active Lipase and Esterase Overexpressed in Heterologous Host.

Organisms/Enzymes	Source	Host	Vector	Localization of Expressed Enzyme	Optimum Temp./Residual Activity	References
*Alkalibacterium* sp. SL3/esterase	Uncultured	*E. coli* BL21 (DE3)	pET-28a (+)	Soluble	30 °C and 68% at 0 °C	[84]
*Chitinophaga pinensis-like*/esterase	Uncultured	*E. coli* RosettaTM (Novagen)	pGEX-6P-2	Insoluble inclusion body	20 °C and NA	[66]
*Lactobacillus plantarum*/*LpLp_2631*/esterase	Microbiological Culture	*E. coli* BL21 (DE3)	pURI3TEV vector	Soluble	20 °C and 90% at 5 °C	[85]
*Burkholderia pyrrocinia*/BpFae esterase	Microbiological Culture	*E. coli* BL21 (DE3)	pET28a pCold-TF and pGEX-4T-1.	Insoluble/soluble non inactive form	NA	[86]
*Candida parapsilosis*/esterase	Cultured	*S*. *cerevisiae*	pYES2	Soluble	NA and at 20 °C	[64]
*Monascus ruber* M7/esterase	Cultured	*E. coli* BL21(DE3)	pET-30a (+)	Soluble	40 °C and 50% at 4–10 °C	[87]
*Alcanivorax dieselolei*/lipase	Cultured	*E. coli* BL21(DE3)	pGEX-6p-1 (GE)	Soluble	20 °C and 95% at 10 °C	[88,89]
*Pseudomonas fluorescens KE38*/lipase	Uncultured	*E. coli* BL21(DE3)	pET28a	Insoluble inclusion body	25 °C and NA	[90]
*Aphanizomenon flos-aquae*/esterase	Uncultured	*E. coli* BL21(DE3)	pET28a	Insoluble inclusion body	5–15 °C	[91]
*Bacillus halodurans*/lipase	Uncultured	*E. coli* BL21 (DE3)	pET-28a (+)	Soluble	30 °C	[92]
*Bacillus licheniformis*/*esterase*	*Cultured*	*E. coli* BL21 (DE3)	pET-28a (+)	Soluble	30 °C and 35% at 0 °C	[63]
*G. antarctica* PI12/esterase	Expressed sequence tag	BL21 (DE3)	pET200_GaDlh	Soluble	10 °C and 50% at 0–30 °C	[93]
*Paenibacillus* sp. R4/esterase	Cultured	BL21 (DE3)	pET-22b (+)	Soluble	35 °C and 45% at 10 °C	[94]
*Pseudomonas* sp./lipase	Uncultured	BL21(DE3)	pET32b (+)	Insoluble inclusion body	35 °C and 50% at 15–40 °C	[27]
*Yarrowia lipolytica*(*LIPY8*)/lipase	Cultured	Insect (Sf9)	pFastBac1	Soluble	17 °C and 70% at 8–30 °C	[65]

NA—not available.

## 4. Purification of Cold-Active Lipolytic Enzymes

Purification is critical in determining an enzyme’s structure and function. Purifying an enzyme not only isolates the target enzyme from other proteins and materials that comprise the crude cell extract but also improves its shelf life and stability. Conformational and structural studies can also be performed after the homogenous purification of the enzymes, and only this homogenous enzyme can be used to establish structure-function relationships [84]. For several decades, protein scientists were into developing screening and optimisation of different combinations of variables during pre-purification and purification experiments Shepard and Tiselius [85] as cited by [86]. The chromatographic pre-purification screening parameters, including resin, ligand, and column screening, are targeted in the experimental design and analytical phases [87]. One example is a high-throughput process development (HTPD) that saves time and cost while harmonising purification procedures through increased automation, miniaturisation, and practical data analysis [88]. A similar format with miniaturised columns enables a high-throughput selection of adsorbent and separation parameters during binding and elution purification experiments. Integrated robot platforms are also employed for choosing a suitable adsorbent in 96-well plates or microcolumn that is essential for determining the success or failure of the purification step [89]. In addition, functionalised microchips, combined with mass spectrometry, are used for protein solution binding, subsequent elution, and analysis. It is possible to determine the optimum binding conditions, the ionic strength for binding, and the lowest ionic strength for the elution [87,90].

Cold-active lipolytic enzymes were purified like other enzymes and proteins sequentially depending on the purity required. For instance, the recommended purity level for structural and functional studies is greater than 98% [91]. Conventional methods include ammonium sulfate precipitation, affinity chromatography, size exclusion (gel filtration), and hydrophobic interaction [84,92,93,94]. Table 2 summarises the various methods used to purify recombinant cold-active lipolytic enzymes. In most cold-active lipase and esterase purification procedures, affinity chromatography is either employed in a one-step or a double-step purification strategy. One-step purification using affinity chromatography generally reduces the time and cost of purification. Even so, the prominent double-step procedure uses ammonium sulfate precipitation with size exclusion and hydrophobic interaction; however, this strategy is suitably employed if the enzymes are produced extracellularly. The affinity chromatography technique is highly specific, while size exclusion, hydrophobic interaction, and ammonium sulphate precipitation are less-specific methods. Sometimes the purpose of using affinity chromatography or ammonium sulphate precipitation in single or first-step purification is to concentrate the recombinant proteins, while less-specific procedures are used to polish the purification. The double-step purification strategy using ammonium sulfate precipitation and nickel affinity has not been utilised much, despite having been reported [95]. In general, obtaining high-purity recombinant enzymes in their stable and active form is expensive, time-consuming, and complex. One-step purification using ammonium sulfate is usually term as partial purification; a well-designed ammonium sulfate precipitation is regarded as a gold standard among several purification strategies [96].

Affinity chromatography is usually achieved by fusing tags at an enzyme’s C or N terminal before its expression [97]. Several affinity tags have been known to facilitate the expression, solubility, detection, and purification of proteins [98,99]. Poly-histidine tagging, also known as His_6_ or His-tag, is widely employed to express and purify most recombinant proteins, including cold-active lipases and esterase [100]. Despite the high affinity, specificity, and size of His-tag, the technique possesses some disadvantages, including (1) co-purification of other histidine-rich microbial host proteins and (2) negative impact on enzyme stability, activity, binding affinity, and structure [101]. The latter is subject to much contrasting opinion and is still debated because some authors observed that its presence is mainly tolerated for enzymes such as lipase; this cannot be ignored due to its effect on reaction specificity. In a study on the thermal stability of some selected proteins conducted by Booth, Schlachter [102], cleavage of the his-tag can be neutral to some of the proteins while influencing the stability of other protein molecules. In general, the his-tag has an effect (positive or negative) or neutral on proteins.

As shown in Table 2, several scholars have reported a single-step purification of cold-active esterase and lipase using nickel Sepharose or agarose affinity chromatography with good fold and recovery. Furthermore, Noby, Saeed [48] have purified a cold-active esterase EstN7 from *Bacillus cohnii* strain with 94.5% yield and 5-fold, adopting Tris–HCl (pH 8.0) in the lysis buffer and potassium phosphate (pH 7.5) in the binding buffer differentiate the study from others that utilised the same buffer in both the purification processes. Kim, Park [103], and Lee, Yoo [104] have purified cold-active esterase using a double-step purification that incorporates nickel-affinity and size exclusion chromatography. Another cold-active lipase, B8W22 from *Bacillus aryabhattii,* was purified in a greater fold of 59.03 using nickel Sepharose affinity and ion-exchange chromatography [105].

**Table 2 ijms-23-15394-t002:** Purification of Cold-adapted Esterase and Lipase.

Enzymes	Type of Purification	Purification Steps	Buffer	Column/Resin	Fold/Yield	Molecular Mass	References
GaDlh	Complete	Single-step/Ni-affinity chromatography	Tris–HCl	Ni–NTA column	1.9/7.7%	28 kDa	[106]
AMBL-20	Partial	Single step/ammonium sulfate precipitation	Tris–HCl	NA	NA	NA	[107]
*Ha*SGNH1	Complete	Single-step/Ni2+-affinity	Tris–HCl	HisTrap HP	2.5/~5 mg/g	24 kDa	[108]
LSK25	Complete	Single-step/Ni-Sepharose affinity	Tris–HCl	Ni Sepharose^®^ 6Fast Flow column	1.3/44%	65 kDa	[27]
AaSGNH1	Complete	Single-step/Ni-Sepharose affinity	Tris–HCl	Ni-NTA agarose	0.6–0.7 mg/mL	43.9 kDa	[109]
B8W22	Complete	Double-step/Ni-Sepharose affinityand ion-exchange	Tris–HCl	DEAE FF column/Octyl Sepharose FF column	59.03/20%	35 kDa	[110]
ERMR1:04	Complete	Triple-step/ammonium sulfate precipitation, Size exclusion, and hydrophobic interaction	Tris–HCl	Sephadex G-100 column, Octyl-Sepharose fast flow column	21.3/NA	250 kDa (hexameric) 39.8 kD (monomeric)	[111]
estHIJ	Complete	Single-step/Ni-affinity	Phosphate buffer	Ni-NTA affinity column.	3.5/47.5%	29 kDa	[112]
ZY124	Complete	Double step/ammonium sulfate precipitation and hydrophobic chromatography	Tris–HCl	Phenyl Sepharose FF column andmicrocolumn reversed-phase LC-1MS	1.34/NA	37.9 kDa.	[105]
AMS8	Complete	Reverse Micelle Extraction	Sodium phosphate	NA	NA/58.84%	NA	[113]
KM12	Complete	Double-step/ammonium sulfate precipitation and ion-exchange	Tris–HCl	Q-Sepharose FF column	15.63/36.0%	33 kDa	[114]
KCTC 22881	Complete	Double-step/affinity chromatography and size-exclusion chromatography	Tris–HCl	HisTrap FF, PD-10 and Sephacryl S200 HR	NA	31.0 kDa	[104]
EstN7	Complete	Single-step/Ni-affinity	Potassium Phosphate	Ni–NTA affinity column	5/94.5%	37.0 kDa	[48]
GlaEst12-like	Complete	Single-step/Ni-sepharose affinity	Sodium Phosphate	Nickel-Sepharose HP	1.7/40%	63 kDa	[115]
RSAP17	Complete	Double-step/ammonium sulfate precipitation and ion-exchange	Tris–HCl	DEAE-cellulose anion exchanger	NA	103.8 kDa	[116]
PsEst3	Complete	Double-step/nickel-affinity and size-exclusion chromatography	Tris–HCl	Ni-affinity and HiLoad 16/60 Superdex 200 column	NA	29 kDa	[103]

NA—not available.

In another double-step purification that used ammonium sulfate and ion-exchange chromatography, Malekabadi, Badoei-dalfard [114] purified a cold-active KM12 from *Bacillus licheniformis using* Q-Sepharose Fast Flow column. Uddin, Roy [116] purified a cold-active RSAP17 from *Ceanisphaera* sp. using a DEAE-cellulose Anion Exchanger. Kumar, Mukhia [111] purified a cold-active ERMR1:04 with 21.3-fold from *Chryseobacterium polytrichastri* by a triple-step with ammonium sulfate precipitation, size-exclusion and hydrophobic interaction using Sephadex G-100 and Octyl-Sepharose Fast Flow columns. Other purification approaches other than conventional approaches were employed for the purification of cold-active lipolytic enzymes, for example, Salleh and Mohamad Ali [113] purified in medium-scale a cold-active AMS8 lipase using reverse micelle extraction (RME) technology. In addition, Zhong, Tian [117] recently purified an esterase Est906 using one-step purification by nucleic acid aptamers with a higher specific activity. Most pre-purification or fractionation steps were done through trial-and-error protocols [118,119].

## 5. Three-Dimensional (3D) Structure and Functional Mechanisms of Cold-Active Lipase and Esterase

Cold-active lipases and esterase have been studied for decades, but few 3D structural data were available for these cold-active lipolytic enzymes. The Crystal three-dimensional (3D) structures are crucial in understanding their biochemical functions toward a cold adaptation. Table 3 summarises the crystal structures of cold-active lipases and esterases. Feller [13] reviewed some experimental methods used in the determination of the psychrophilic enzymes crystal structures and reported that only one structure was determined using NMR: most of the published crystal structures utilised X-ray diffraction as their experimental method. For years this has stayed the same compared to their mesophilic and thermophilic counterparts.

The studies on cold-active lipase and esterase were not limited to the isolation and characterisation of these novel enzymes, but also developed a theoretical model regarding their low-temperature adaptation mechanism. The need to establish the specific features that aid their catalytic functions at low temperatures compared to their mesophilic and thermophilic counterparts makes it necessary to analyse the available data on these lipolytic enzymes. The general catalytic mechanism of lipase and esterase is that of serine hydrolases enzyme that involves a nucleophilic attack on the substrate during the acylation step, which forms a covalent complex of enzyme and substrate, followed by the diacylation step in which the enzyme-substrate complex is hydrolysed by a molecule of water [128,129].

The mechanism of the transesterification reaction of lipases is similar to their hydrolysis reaction mechanism as reviewed by Jegannathan, Abang [130]; the biocatalytic process involves a catalytic triad that serves as a charge-relay system, followed by the creation of an oxyanion hole and formation of tetrahedral intermediates. The catalytic triad of lipases and esterase is highly conserved regardless of whether they are of mesophilic, thermophilic, or psychrophilic origin [131,132]. Therefore, the focus is not on how they catalysed their reaction but on how they performed it in low temperatures in the case of cold-active enzymes. The mechanisms of psychrophilic protein adaptation have been widely reviewed [9,133,134].

The higher local (localised to the catalytic regions) and global dynamics of cold-active enzymes allow them to act in a more disordered lowest energy state [1,135]. De Maayer, Anderson [1], described the structural modifications such as extended surface loops, increased mobility and glycine clustering at the catalytic site, and increased number and size of enzyme cavities were common in cold-active enzymes where they increased their specific activities and flexibilities while decreasing their thermal stability. Hashim, Mahadi [106] further demonstrated that the cold-active esterase-like exhibits several properties of cold-adapted enzymes, such as glycine clustering in the binding pocket, low hydrophobicity of the enzyme core, and the lack of proline residues in the loops. Noby, Auhim [135] described the dominant cold adaptation mechanism as likely to be dealing with two independent mechanisms: the tolerance to changes in water entropy, which is low in the solid phase and higher in the gaseous phase [136]. Water molecules will be more ordered and viscous as the temperature drops, diminishing the hydrophobic effect essential for keeping protein in its folded state [137]. An increased surface negative charge is thought to play a role in addressing water entropy through the retention of stable hydrophobic interactions by increasing the interactions of surface residues with water, despite fluctuations in entropy and viscosity [137]. Adjustment to shift in water entropy of cold-active enzymes has been postulated earlier [138,139]. In esterase with an active site located at the end of molecular tunnels, it was noticed that cold activity was related to improved substrate accessibility to the active site by forming additional tunnels to access the active site and increasing the volume of the active-site cavity. This was noticed by comparing cold-active esterases with other mesophilic or thermophilic closest homologues [140,141].

In contrast to the well-established notion that metal ions hinder the structural flexibility of enzymes [142], a study revealed that metal ions, either directly or indirectly, contribute to the improvement of the cold activity of a psychrophilic enzyme. The enzyme’s active site had two manganese ions (Mn^2+^-Mn^2+^) with a significant weak exchange coupling in the absence of a substrate, which rearranged and formed a well-tuned structure upon substrate binding. The di-Mn^2+^ ions maintained the ‘loose’ structure responsible for keeping the enzyme active site flexible and further enhanced its performance at low temperatures [143]. Another role of Manganese Mn^2+^ on the low-temperature adaption of a cold-active esterase was recently described by Marchetti, Orlando [144]; as with other psychrotolerant and psychrophilic homologues, the Mn^2+^ binding site was discovered on the surface of the enzyme close to the active region and the esterase’s interaction with the Mn^2+^ ion only results in a local conformational shift near its active site, which unexpectedly improves both its catalytic efficiency and thermal stability [144].

In a recent study on the structural basis of cold-adaptation of two orthologous mesophilic-psychrophilic bacterial lipases, van der Ent, Lund [128] observed a limited number of mutations (34 out of 181 residues) that were responsible for their thermal adaptation. Only single amino acid was found close to the substrate binding site, and the remaining mutations were found farther away on the enzyme surface. They further suggest that a combined effect of the mutations might likely change the activation enthalpy and entropy as in other cold-adapted enzymes. Further experiments, such as more crystal structures, functional studies, and effective computer simulations, are needed to unveil different novel cold-adaptation strategies. While investigating the origins of enzyme functions through the sequence, structure, and reaction mechanism, Furnham, Dawson [145] made the surprising discovery that a large number of enzyme domain superfamilies share at least one catalytic residue, which suggests that enzyme functions have originated from a common ancestor with generic functionalities. Rizzello, Romano [146] identified a specific area of seven amino acids contributing to cold adaptation. Therefore, knowledge of evolutionary traits such as domain or motif sharing between other cold-active enzymes from the same organism could also answer their cold adaptation. The specific cold adaptation process of cold-active enzymes, such as lipases and esterases, needs to be better understood. To adapt to low temperatures, cold-active enzymes use a combination of strategies, some of which might have unintended consequences during enzyme evolution. Although several cold-adaptation techniques have been identified, there is still much more to learn about how organisms adapt to the cold.

## 6. Conclusions and Future Perspectives

Despite their unique characteristics and enormous potential applications of cold-active enzymes, there are still obstacles from laboratory to large-scale industrial applications. In this review article, we have examined cold-active lipases and esterases that have been studied primarily from 2018 to the present, focusing on their recombinant overexpression, purification, three-dimensional structural elucidation, and molecular mechanism towards cold adaptation, which has recently not been reviewed otherwise, although most of the areas discussed were not specific to cold-active esterases and lipases, but still relevant. The lack of universal analyses as the status quo due to the dynamic nature of proteins is the greatest challenge facing separation and purification aspects. Focusing on a quick and efficient purification process will increase 3D structure elucidations quickly to improve our understanding of this cold-active lipolytic enzyme. We could not answer how purification relates to the cold activity of lipase and esterase. Previous studies have shown that cold-adaptation processes of cold-active enzymes, such as lipases and esterases, do not indicate any directional trend; a wide range of solutions evolved, during enzyme evolution, some of which had counterproductive consequences such as activity-stability trade-offs [147] characterised by increase cold activity with consequent poor stability. The resolved crystal structures were reviewed in Table 3. This gap is only very slowly being filled. This is expected to significantly impact understanding nearly all aspects of enzyme function, such as stability, catalysis, substrate binding, and regulation.

## Figures and Tables

**Figure 1 ijms-23-15394-f001:**
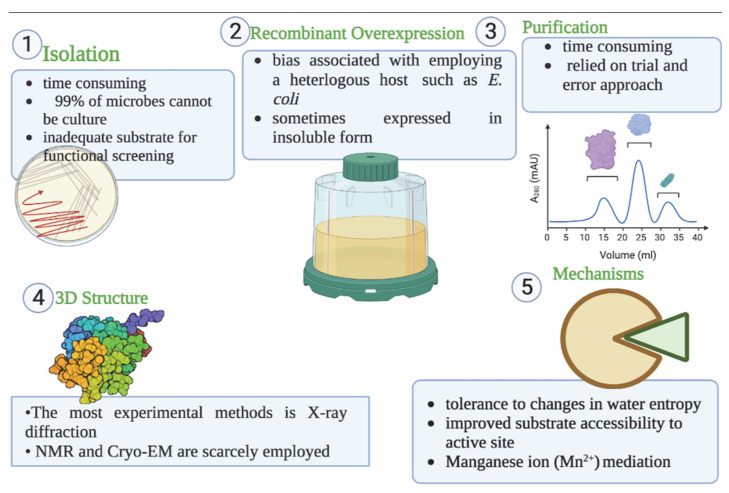
Summary of various sections of this review (Created with BioRender.com).

**Table 3 ijms-23-15394-t003:** Summary of resolved crystal structures of cold-active lipases and esterase.

Enzymes	PDB Code	Organism	Expression System	Experimental Method	Resolution (Å)	Ligand	References
Esterase	4V2I	*Thalassospira* sp.	*Escherichia coli* BL21(DE3)	X-ray Diffraction	1.69	Magnesium ion	[50]
Esterase	4AO8	Arctic Intertidal Metagenomic Library.	*Escherichia coli* K-12	X-ray Diffraction	1.61	Dihydroxyethyl Ester	[120]
Esterase	5DWD	*Pelagibacterium halotolerans* PE8	*Escherichia coli*	X-ray Diffraction	1.66	2-(2-{2-[2-(2-Methoxy-Ethoxy)-Eth0xy]-Ethoxy}-Ethoxy)-Ethanol	[121]
Esterase	3I6Y	*Oleispira antarctica*	*Escherichia coli* BL21(DE3)	X-ray Diffraction	1.75	Dihydroxyethyl Ester	[122]
Lipase	6ISP	Laboratory Evolution of *Moesziomyces antarcticus*	*Escherichia coli* BL21(DE3)	X-ray Diffraction	1.88	N, N-Bis(3-D-Gluconamidopropyl) Deoxycholamide and Calcium Ion	[123]
Lipase	6ISR	Laboratory Evolution of *Moesziomyces antarcticus*	*Escherichia coli* BL21(DE3)	X-ray Diffraction	2.60	Tetraethylene Glycol	[123]
Lipase	6ISQ	Laboratory Evolution of *Moesziomyces antarcticus*	*Escherichia coli* BL21(DE3)	X-ray Diffraction	1.86	1,2-Ethanediol	[123]
Lipase	5GV5	*Moesziomyces antarcticus*	*Aspergillus niger*	X-ray Diffraction	2.89	[(1s)-2-(Methoxycarbonylamino)-1-Phenyl-Ethoxy]-Propyl-Phosphinic Acid	[124]
Lipase	5A6V	*Moesziomyces antarcticus*	*Aspergillus oryzae*	X-ray Diffraction	2.28	Xenon	[125]
Lipase	5A71	*Moesziomyces antarcticus*	*Aspergillus oryzae*	X-ray Diffraction	0.91	Isopropyl alcohol	[125]
Lipase	5CH8	*Penicillium cyclopium*	*Komagataella pastoris*	X-ray Diffraction	1.62	Glycerol	[126]
Esterase	7B1X	uncultured bacterium	*Escherichia coli*	X-ray Diffraction	2.30	None	[36]
Esterase	7DDY	*Arcticibacterium luteifluviistationis*	*Escherichia coli* BL21(DE3)	X-ray Diffraction	2.50	None	[127]
EsteraseD	6JZL	*Shewanella frigidimarina*	*Escherichia coli*	X-ray Diffraction	2.32	None	[40]

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
