# Peer review of "Cold-Active Lipases and Esterases: A Review on Recombinant Overexpression and Other Essential Issues"

_ijms, 2022, doi:10.3390/ijms232315394_

Round 1
Reviewer 1 Report
Overall, this is a nicely written review covering a range of different aspects for cold-adaptive lipases. However, I feel the review could be further improved if in-depth discussion on the structure-activity relationship, and how mutating certain residues affect enzyme activity at cold temperature.
A few recommendations:
- Section 2: “Approaches for isolation of genes encoding cold-active lipase and esterase”
A table providing a host, method of isolation, and working temperature would be great for reader to follow
- Section 3: “Recombinant expression…”
I feel this section lack in-depth discussion. Authors just simply provide a table listing out gene being expressed, predominantly in E. coli, except for one example in yeast S. cerevisiae.
Authors also did not explain why certain lipases can be expressed, and some are not. For examples, lipases from gram negative bacteria such as Pseudomonas or Burkholderia are known to be difficult to be expressed due to requirement of native host chaperon protein. Authors should make attempt to review those paper that tackle issue of insoluble expressed proteins in recombinant host.
Furthermore, more examples in other heterologous host such as Pichia, Bacillus, insect cells, etc would be great addition, rather than just E. coli.
- Section 4 is missing ?
- Section 5:
Would be nice to explain why hydrophobic interaction or ion-exchange was used instead of standard His-tag method for purification of some of the lipases. Are his-tag cannot be used in these cases or what the limitation?
Hydrophobic interaction or ion-exchange have lower selectivity than His-tag and normally only used when His-tag fail. An explanation to rationale why certain purification method was chosen would be benefit to the general reader
- Section 6:
“Noby, Johnson [116] described the prime cold adaptation mechanism as likely to be dealing with change in water entropy that allowed improved access for the substrate to the active site”
I think the author misunderstand the mechanism of cold-adapted esterase described by Noby et al.
There are 2 independent mechanism int Noby et al work: “Thus, it is likely that (1) increased substrate accessibility and (2) tolerance to changes in water entropy are the main drivers of EstN7’s cold adaptation rather than changes in dynamics. “
(1) And (2) are unrelated mechanism, thus by saying “change in water entropy that allowed improved access for the substrate to the active site” is incorrect, as change in water entropy did not lead to an improved access for the substrate.
Overall, I feel this section need more in-depth discussion.
Author Response
Comment #1:
-- Section 2: “Approaches for isolation of genes encoding cold-active lipase and esterase”
A table providing a host, method of isolation, and working temperature would be great for reader to follow
Response 1:
Thank you for the suggestion.
Comment #2:
-Section 3: “Recombinant expression…”
I feel this section lack in-depth discussion. Authors just simply provide a table listing out gene being expressed, predominantly in E. coli, except for one example in yeast S. cerevisiae.
Response 2:
The section was reconstructed and updated in the revised manuscript
Comment #3:
- Authors also did not explain why certain lipases can be expressed, and some are not. For examples, lipases from gram negative bacteria such as Pseudomonas or Burkholderia are known to be to be expressed due to requirement of native host chaperon protein. Authors should make attempt to review those paper that tackle issue of insoluble expressed proteins in recombinant host.
Furthermore, more examples in other heterologous host such as Pichia, Bacillus, insect cells, etc would be great addition, rather than just E. coli
Response 3:
The paragraph was reconstructed for more clarity and updated in the revised manuscript
Comment #4:
-Section 4 is missing ?
Response 4:
The missing section was corrected and updated in the revised manuscript
Comment #5:
Section 5:Would be nice to explain why hydrophobic interaction or ionexchange was used instead of standard His-tag method for purification of some of the lipases. Are his-tag cannot be used in these cases or what the limitation?
Hydrophobicinteraction or ion-exchange have lower selectivity than His-tag and normally only used when His-tag fail. An explanation to rationale why certain purification method was chosen would be benefit to the general reader
Response 5:
Thank you for this suggestion. The explanation was added and updated in the revised manuscript
Comment #6:
Section 6: “Noby, Johnson [116] described the prime cold adaptation mechanism as likely to be dealing with change in water entropy that allowed improved access for the substrate to the active site”
I think the author misunderstand the mechanism of cold-adapted esterase described by Noby et al.
There are 2 independent mechanism int Noby et al work: “Thus, it is likely that (1) increased substrate accessibility and (2) tolerance to changes in water entropy are the main drivers of EstN7’s cold adaptation rather than changes in dynamics. “(1) And (2) are unrelated mechanism, thus by saying “change in water entropy that allowed improved access for the substrate to the active site” is incorrect, as change in water entropy did not lead to an improved access for the substrate.
Overall, I feel this section need more in-depth discussion
Response 6:
Thank you for the comment. Yes, Noby et al proposed two independent mechanisms. The section was improved and updated in the revised manuscript.
Reviewer 2 Report
The Review entitled “Cold-Active Lipases and Esterases: a review on cold adaptation mechanisms and some neglected issues” by Matinja et al. wants to report the molecular mechanisms of cold-adaptation, purification, three-dimensional structure, overexpression, and biotechnological application of cold-active esterases and lipases (listed in the order they appear in the abstract). The review has five major drawbacks, that prevents it to be considered by the Journal: i. The used terminology, the syntax and logical order of some sentences is very poor and inadequate even for a master thesis, preventing this reviewer from understanding part of the contents; ii. The title is not adequate for the content of the review, which, as stated by the authors in their abstract, is not focused on the molecular mechanisms (reported in just a paragraph) and neglected issues, but report the molecular mechanisms of cold-adaptation, purification, three-dimensional structure, overexpression, and biotechnological application of cold-active esterases and lipases; iii. All the reported topics can be found in previous reviews, while this work treats the topic in a very superficial manner and is far from being complete; iv. Most of the topics are not specific to cold-active esterases and lipases, and can be generalized to all cold-active (if not all) enzymes in general; v. Some arguments and definitions are wrong, phenomenalist or non-sensical from a scientific point of view.
In the attachment, there is a list of points to revise, which can be useful for fixing the actual content and formal issues; anyway, a more in depth reading of the cited literature is suggested, to fetch out more contents and select important and recent information on molecular mechanism of cold-adaptation in cold-active esterases and lipases.

Author Response
We appreciate the time and effort and dedication that you put on our manuscript, you indeed provide valuable feedback and constructive suggestions.
Please, find below our responses to the reviewer’s comments:
- Lines 16-17: better to rephrase as "to perform an efficient metabolic flux at low temperatures, necessary to stay alive and reproduce in those conditions".
Response 1: Thank you for the suggestion. The sentence was rephrased as “to perform an efficient metabolic flux at low temperatures, necessary to thrive and reproduce under those constraints” and changed in the revised manuscript.
- Line 18: what is the definition of an ideal catalysts for the authors?
I agree about listed advantages, but in most of the cases cold-active enzymes cannot grant the same catalytic efficiency of mesophilic or thermophilic enzymes, and it may be economically better (time, money and environment speaking) to spend money for heating + 30°C and have a process three times faster or more. As the review does not clarify at this point in the manuscript, I suggest the authors to avoid using the term "ideal" for the listed properties. Otherwise, they should provide examples of processes that can be done both at 25°C or 55°C and indicate that the total energy consumed (heating*time) is lower at 25°C then 55°C.
Response 2:
Thank you for the comment. Perhaps, the term “ideal” used is not as in perfect, supreme, or most suitable, instead it is used in the case of “ideal world” which refers to desirable, visionary, or perfect but is not likely to become a reality.
This is in line with what you have indicated, spending on heating to achieve good results makes thermophiles and mesophiles more acceptable enzymes. This is consistent with the goal of our review, to examine the development of psychrophiles after a certain period. However, the sentence has been upgraded in the revised manuscript as “Cold-active enzymes are ideal biocatalysts that can reduce the need for heating procedures…….”
- Line 20: authors can claim that it is interesting to provide an update on the cold-adaptation mechanism of cold-active enzymes, but this is not related to this statement: it is remarkable that most of the processes done with the use of enzymes are economically better to be done at > 50°C, while cold-active enzymes are best suited for processes that require constantly cold-temperatures or can provide a qualitatively superior product or need a fast and cost-effective inactivation procedure. Nothing reported in this review makes the reader convinced of something different from that.
Response 3: Thank you for the comment. Currently, enzymes are economically wise to be conducted at high temperature. The explanation could be the difficulty in adapting technologies developed at the laboratory scale to the industrial level due to the extensive research and development that is required, which psychrophilic enzymes are currently facing. But with more in-depth research, things might turn around in the future.
- Line 34: you should write the introduction indicating what stands for "psychrophilic environment". Is what is described above? If so, please indicate that the previous description is meant for a psychrophilic environment.
Response 4: The issue was addressed in line ~30 of the revised manuscript.
- Line 35: “Archea” and “Bacteria” should be uppercase. “Eukaryote” is not a taxonomic rank: use "eukaryotes" or "Eukarya" instead.
Response 5: The corrections were made and updated in the revised manuscript.
- Line 36: rewrite: "As cellular activities are ..." Moreover: why cellular activities are disrupted at cold temperatures? Please say it in two lines.
Response 6: The paragraph reconstructed and updated in the revised manuscript.
- Line 39: not clear. Organisms can adapt to thrive in cold-environments by controlling the nucleation of small ice-crystals outside their cells, but here the concept is not clearly expressed.
The sentence has poor construction.
Response 7: The sentence was reconstructed and updated in the revised manuscript.
- Line 41: the survival strategy is “for cold-adapted environment”.
Response 8: The sentence was corrected and updated in the revised manuscript.
- Line 42-43: same as lines 16-17.
Response 9: The sentence was corrected and updated in the revised manuscript.
- Lines 44-45: avoid repeating "enzymes" in the same statement. Moreover, "high catalytic activity" must be declared immediately that is with respect to mesophilic and thermophilic enzymes, as said in the next sentence.
Response 10: The sentence was corrected and updated in the revised manuscript.
- Lines 48-50: simplify this complex statement. Moreover, the subject of this statement are "structures", but later in the statement it is used a singular pronoun. What is the "polypeptide's spatial structure"?
Response 11: The syntax was corrected and updated in the revised manuscript. The spatial polypeptide structure is referring to the quaternary structure of proteins.
- Line 51: how could this be possible if in the next statement it is indicated that there are weak interactions? As far as I know, it is the opposite, they have low-affinity and tend to be more promiscuous.
Response 12: Agreed. The statement was corrected and updated in the revised manuscript.
- Line 52: this statement must be said if compared to mesophilic or thermophilic homologues.
Response 13: The statement was added and updated in the revised manuscript.
- Lines 53-54: "… the high catalytic efficiency at low temperature…"
Response 14: The sentence was corrected and updated in the revised manuscript.
- - Line 56: Which transformations? Please list them.
Response 15: The transformation is removed for clarity, instead example of bio-catalysis is mentioned and updated in the revised manuscript.
- Lines 56-57: What a hydrolase is? This is an introduction section, so please describe better and provide a ref.; I cannot agree with the statement “cold-active hydrolases were the most frequent enzymes characterized and used for industrial purposes”.
Response 16: Absolutely. The comparison is between the cold-active hydrolases and other cold active enzymes. The sentence was corrected and updated in the revised manuscript.
- Lines 59-60: Poor phrase construction and mixing of different concepts: one thing is the reduced heating/energy consumption, another the quality of industrial production.
Response 17: The phrase was reconstructed and updated in the revised manuscript.
- Lines 63-64: Confusing statement:? a structural protein fold and a protein family are different concepts.
Response 18: they are two different concepts. But the basis for serine hydrolase superfamily is the α/β hydrolase fold. The sentence was rephrased and updated in the revised manuscript.
- Line 66: esterases cannot work in the synthesis direction? What simple means? Please use a scientific language (i.e. short chain). Once you defined as simple the term "short-chain", then you can use this term for this property.
Response 19: The esterase is also working in the synthesis direction. The term ‘synthesis’ and ‘short chain’ were updated in the revised manuscript.
- Lines 68-69: Which organic solvents? ; please fix the use different verb tenses in the same statement.
Response 20: The statement was corrected and updated in the revised manuscript
- Line 72: Use past tenses, you are not speaking of the present.
Response 21: Corrected and updated in the revised manuscript.
- Lines 72-73: “In recent years, there is a significant rise in the number of articles published on cold-active lipases and esterase.” Please count them or provide a reference that reported this rise.
Response 22: The statement was and updated in the revised manuscript.
- Line 73: now are you repeating concepts of the cold-active enzymes or speaking about coldactive lipases? It is not clear.
Response 23: We try to focused on cold-active lipase and esterase, which means only papers with this keyword were examined. However, most of the issues are not specific to the cold active lipase and esterase, but sometimes to cold active enzymes or proteins in general.
- Lines 75-76: most previous studies focused on the isolation of cold-active lipases and esterase from various sources, their overexpression, and biochemical characterization. Please, provide a couple of refs.
Response 24: Done and updated in the revised manuscript
- Lines 78-82: this is better if reported in the order that is shown in Fig 1. Figure 1 is not related to just the production issues of cold-active lipolytic enzymes, as indicated in the caption. It is related to the characterization of enzymes, in general, regardless they are lipolytic or do whatever else. Moreover, the figure utility is questionable, as the text is not clear, and the "setback" and "possible remedy" are listed only for the "Screening" step.
Some examples of problems with the figure are:
- why speaking about "utilizing synthetic biology approaches" if never introduced at this point of the review?
- all the issues stated, except the mechanisms, can be generalized to any enzyme that is taken from a natural organism.
- what is the "bias associated with employing a heterologous host?". Why should protein folding involved with overexpression? It should be related to its solubility.
- why is protein purification expensive? I mean, once an enzyme is soluble in the condition of expression, why should be it difficult to purify? The purification of an enzyme is usually not a problem once it is overexpressed in a soluble form.
- the 4) point is a general point that concerns structural biology in general and is related to the differences, technical problems, and costs of the different techniques. Moreover, the problem of monomer in solution being a dimer in the crystal is a universal artifact that may arose for some proteins, not just psychrophilic enzymes (please remember you stated this review is focused on esterases and lipases).
- same as above, you focused on lipolytic enzymes, not all enzymes. Why should you report one of the last cold-adaptation mechanism in this figure? Why do you explain this mechanism in such a poor way? The statement is also poorly written (i. e., "accessed" instead of "access").
Response 25: The Figure was corrected and updated in the revised manuscript
- Line 95: "B" does not stand for a type of strain, but to the evolutionary group of lipases to which CALB belong to. Remove it. Which “other few enzymes”? It could be very useful if which ones are employed in the industries are reported in a table.
Response 26: Rephrased and updated in the revised manuscript
- Line 96: you should describe the method here instead later in the review.
- Lines 98-99: considering this is a review, you should briefly describe the steps that you listed here.
Response 28: The steps were briefly described and updated in the revised manuscript.
- Lines 100-102: bad sentence construction.
Response 29: The sentence was reconstructed and updated in the revised manuscript
- Line 102: use past verb tenses to speak about past work.
Response 30: The verb tense was corrected and updated in the revised manuscript.
- Lines 106-107: this statement is valid for all enzymes, not just lipases and esterases, so it should be reported in the introduction section.
Response 31: Done. The statement is reported in the introduction section and updated in the revised manuscript.
- Lines 108-110: incorrect: metagenomics and microbial genomics are two different concepts: metagenomics refers to environmental DNA that can be sequenced and annotated by bioinformatic tools; microbial genomics can be done only after culturing an isolated microbial species. Which experimental methods are you referring to? Which bioinformatics tools?
Response 32: The lines were corrected and updated in the revised manuscript. The experimental methods include high-throughput sequencing such as Illumina, and the bioinformatics tools were various database and software’s that can be used or covered sequencing, metagenomic assembly, binning, domain prediction, targeted gene discovery, pathway databases, and data sharing.
- Line 111: use either the term "enzyme" or "biocatalyst", they mean the same.
Response 33: The term “biocatalyst” was removed and updated in the revised manuscript.
- Lines 114-115: not clear. What do you mean for genetic information on enzyme functions?
Response 34: It was an oversight, the term “genetic” is not supposed to be in the sentence, it is therefore removed and updated in the revised manuscript.
- Line 119: what is a metagenomic library?
Response 35: “Metagenomic libraries are DNA fragments extracted from environmental or community samples and cloned into specific vectors, the smaller fragments that are less than 15Kb are constructed into plasmids, whereas larger inserts (25 to 200Kb) are constructed in vectors such as cosmids and fosmids.” The definition is added to the revised manuscript.
- Lines 121-122: please explicit what are the reasons for the "vibrant and dynamic nature" referred to the metagenomic approach.
Response 36: the vibrant is referring to the expansion of our understanding of the evolution, ecology, diversity, and function of microbial world.
- Line 124: I do not agree with this statement. Metagenomics is in use since decades and not in an early state. It is true there are challenges, such as any other technique, but this is not due to the infancy of the approach.
Response 37: The statement on its infancy was removed and updated in the revised manuscript.
- Lines 125-132: bad and complex sentence construction, make it difficult to understand the content. Introns are present only in Eukarya, not in Bacteria, that are the microbes the authors were speaking on in the previous lines. The reported problems are not a limit of the metagenomic approach, but of the enzyme discovery in general. “adequate substrate scarcity for functional screening” is a non-sensical statement, the substrate is adequate or scarse, not both.
Response 38:
The sentence was rephrased for more clarity and updated in the revised manuscript. The word “adequate” was wrongly inserted. Metagenomics as a tool for enzyme discovery, it shared some of the challenges of the enzyme discovery. The presence of fewer genes encoding enzymes of interest in metagenomic DNA reported by Chopra et al is sometimes because of introns in case of some fungi although their introns are shorter than that of mammalian.
- Line 133: what do you mean for "bias" in this context?
Response 39: One example of bias in using a heterologous host such as E. coli is in the case of signal peptide expression of fungal enzyme, a situation that sometimes destabilized the enzyme thermodynamic, reduces its activity or even increases its aggregation.
- Lines 135-136: bad use of verb “discovered”.
Response 40: The verb is changed and updated in the revised manuscript
- Lines 139-140: how is this related with the title of the paragraph?
Response 41: The paragraph was reconstructed and updated in the revised manuscript
- Line 143: many enzymes, not just psychrophilic, have solubility issues. The expression yield in terms of mg/L culture should be reported, not just a generic indication of solubility or not, as different level of solubility can be found.
Response 42: The statement was rephrased and updated in the revised manuscript. Unfortunately, none of the literature examined indicate the expression yield in terms of mg/L.
- Lines 144-146: this statement is not clear: how structural and spectroscopic studies should be related to biotech and industrial applications? I can do such studies also on a structural protein, without any enzymatic activity.
Response 43: The statement was removed and updated in revised manuscript
- Lines 146-148: this statement should be explained better. Why should someone be interested in expressing an enzyme in the cold-adapted natural host?
Response 44: The statement was corrected and updated in the revised manuscript.
- Line 150: it should have been the 4th paragraph, not 5th.
Response 45: The numbering was corrected and updated in the revised manuscript.
- Line 157: what is intended with the needed purity? 70%, 80%, more?
Response 46: The purity needed was added and updated in the revised manuscript.
- Lines 158-161: poor phrase construction. Centrifugation, filtration, and dialysis are usually part of those previously mentioned.
Response 47: The phrase was reconstructed and updated in the revised manuscript.
- Lines 166-168: this information is meaningless without a quantitative analysis of the literature with respect to not cold-active lipases and esterase. It is likely that this two-step strategy is not used too much for any enzyme.
Response 48: The statement was improved and updated in the revised manuscript
- Lines 170-172: this statement is non-sensical. In general, in this paragraph it is not reported clearly what are the advantages and disadvantages of listed protein purification techniques, and, most importantly, how is this related with cold-activity and esterase/lipase.
- Lines 178-179: it is widely employed for proteins in general, regardless they are enzymes or not.
Response 50: The statement was corrected and updated in the revised manuscript
-Lines 182-184: please argument this statement at least by citing some literature, as I do not agree with it.
Response 51: The statement was updated in the revised manuscript
- Lines 189: this is the same as the affinity chromatography previously mentioned. Moreover, this method is called in different ways throughout this paragraph, making hardly difficult to follow and understand.
Response 52: It is the same affinity chromatography mentioned earlier. They are called in different ways in other to give the reader a clue on which type of affinity chromatography is being used in a particular study.
- Lines 191-193: what is the difference of these two treatments? Are the studies working on the same proteins? If not, they are not comparable.
Response 53: The statement is not trying to compare between two treatments, instead to pinpoint that the lysis buffer and binding buffer can be different during purification. This will make an impact on selection of suitable buffer lysis and purifications.
- Line 198: this is not understandable.
Response 54: The sentence was rephrased and updated in the revised manuscript
- Lines 208-213: the utility of this list of purification methods is questionable, as it should be necessary to compare different methods for the same protein to know if one or more of these approaches are more problematic for cold-active lipases or esterases. Moreover, is the activity of the protein related to the purification method under use? Explain better, as it is not clear –
- Lines 215-217: bad sentence construction and it is not clear why how this related to coldactive lipase and esterase.
Response 56: The sentence was readjusted.
- Lines 230-231: how ice-binding proteins are related to cold-active esterases and lipases?
Response 57: The statement was removed and updated in the revised manuscript
- Lines 231-234: this ice-binding purification works for ice-binding proteins because they bind to ice surface. While it was shown how some proteins gain an ice-binding function during their molecular evolution, this usually pass through their inactivation and loss of enzymatic activity, and no evidence of active enzymes nor esterase/lipase binding ice was reported. Therefore, this comment is non-sensical.
Response 58: The comment is retracted and updated in the revised manuscript
- Line 236: the information reported here is not related to enzyme purification and is wrong. Cryo-EM is not able to increase protein purification yield, not even AI was used for developing better protocols. AlphaFold has an accuracy of ≈85%.
Response 59: The paragraph is removed for more focus
- Line 237: this statement is phenomenalism and not scientific.
Response 60: The statement was removed and updated in the revised manuscript
- Lines 239-241: the conclusion of the paragraph is not connected with the rest of it.
Response 61: The paragraph was rephrased in the revised manuscript
- Paragraph 6.1 is confusing and not specific to cold-active lipases and esterases.
Response 62: The statement was updated in the revised manuscript
- Line 255: Why? The oligomerization state can be easily determined experimentally once soluble proteins have been purified, allowing to determine any oligomerization artifact due to experimental methods for 3d structure solution.
Response 63: The statement was removed. Previously, we wanted to explain how the oligomeric state affect their temperature adaptability.
Lines 268-270: bad syntax.
Response 64: The syntax was corrected and updated in the revised manuscript
- Most of the 6.2 paragraph describes the general mechanism of catalysis, instead providing clues on cold-adaptation mechanism of esterases and lipases.
Response 65: The paragraph was restructure and explained in the introduction section.
- Lines 275-276: this is just providing general information on the catalytic mechanism of lipase and esterase, not on cold-adaptation mechanisms.
- Lines 280-282: incomplete and poorly described mechanism. Moreover, here it is not presented how the transesterification reaction happens.
Response 67: The information on their cold-adaptation mechanism was scanty across the articles examined for this review, the reason could be that the enzymes can applied without fully knowing its cold adaptation mechanism and maybe the reason why only fewer articles reported this aspect. Their reaction for hydrolysis, esterification and transesterification are similar, in such a way they favour a reaction based on available substrates and it is applicable to all lipase and esterase irrespective they are of psychrophilic, mesophilic, or thermophilic origin. In addition, only few report captured their low temperature adaptation mechanism. The issue is now clearly stated in the last paragraph of the introduction section.
- Line 286: The higher specific activity is not absolute, but related to reaction carried out at low temperatures. Moreover, such features have already been presented in the introduction section, so must be removed from this part.
Response 68: The statement was removed and updated in the revised manuscript
- Line 290: The meaning of this statement is not clear. What are "local" and "global" dynamics?
Dynamics of what?
Response 69: The statement was rephrased for clarity and updated in the revised manuscript
- Line 291: please, list these structural modifications.
Response 70: The structural modifications were added and updated in the revised manuscript.
- Lines 296-299: Tryptophan residue is bulky and have a big steric hindrance. How can this be coupled with this description of the mechanism? This statement is not clear.
- Line 300: is this statement referred to the same enzyme of the previous one? What is the plug? How can high substrate specificity being considered a cold-adaptation mechanism, as it is known that cold-adaptation usually comes along with a higher substrate promiscuity?
Response 72: Thank you for pointing it out and we agree with it. It was the mutation of Tryptophan to Alanine that produce this effect of eliminating the steric hindrance. Further the statement is not related to the cold adaption but instead the substrate specificity. We therefore remove the statement from the revised manuscript.
- Line 302: what "prime cold adaptation mechanism" stands for?
Response 73: “ dominant “ according Noby et al
- Lines 304-305: Very poor sentence construction. This reviewer was not able to understand the meaning of this sentence.
Response 74: The sentence was reconstructed and updated in the revised manuscript.
- Line 308: this statement has no scientific meaning.
Response 75: the statement has been improved and updated in the revised manuscript
Line 309: "acetylation" and "diacylation" are wrong terms, "acylation and" "deacylation" are the right terms.
Response 76: The terms were corrected and updated in the revised manuscript.
- Lines 310-311: general description that does not mean anything.
Response 77: The description is trying to recap on the mechanisms of actions of transesterification and hydrolysis of lipase and esterase. The general mechanism is not limited to cold active lipase or esterase, but it also includes those of mesophilic and thermophilic origin.
- Lines 314-316: this statement has poor construction, is syntactically wrong and meaningless in content.
Response 78: the statement has been improved and updated in the revised manuscript
- Lines 319-320: "survival mechanisms" is an improper terminology for the cold-adaptation of enzymes, which are not living beings
Response 79: the terminology was corrected and updated in the revised manuscript.
- Lines 322-324: Why such strategy is proposed? Is there any known evidence of domains that have the function to increase the cold-activity of enzymes? I think this perspective has no specific motivation.
Response 80: The example of domain that increase cold-adaptation were added and updated in the revised manuscript
- Line 327: which substrates? Please, make a table to list them
- Lines 329-330: I think mesophilic lipases are more promising for such applications, given the temperature of the reaction. Moreover, the description of this application is insufficient and the reader cannot get an idea of what is this about
- Lines 330-333: These two statements seem contrasting in meaning. Are the enzymes listed in table 4 applied or not? If yes, why the second statement indicates that research on their application at cold temperatures is lacking? You meant that these enzymes are cold-active, but used at higher temperatures for those applications? Please be clearer and more explicit.
Response 83: The section is removed to focus on other topics.
- - Line 340: "physiology" is an improper term for enzymes. Why is knowledge on the molecular mechanism of cold-adaptation useful for applying such enzymes? If I have a cold-active enzyme that is useful for an application, it can be applied regardless the knowledge on its property. The given explanation is not convincing.
Response 84: The statement was rephrased and updated in the revised manuscript.
- Line 344: please avoid the use of such phenomenalist terms in a scientific work
Response 85: The terms was removed and updated in the revised manuscript.
- Lines 344-347: poor statement construction and meaningless content. Which is the desired purity? Was never mentioned in the main text. Moreover, this statement can be given for any enzyme, not just cold-active esterases or lipases.
Response 86: The statement was removed and updated in the revised manuscript.
- Lines 350-352: the cold adaptation mechanisms of recently studied lipases and esterases were not explicitly presented in this review. The considerations made for expression, purification and structure determination were presented in a superficial and confusing manner; moreover, they are not neglected areas at all, and not specific for cold-active esterases and lipases.
Response 87: The sentence was rephrased and updated in the revised manuscript.
- Line 353: avoid presenting new things in the conclusion section.
Response 88: The sentence was corrected and updated in the revised manuscript.
- Lines 354-355: “this is expected to have a major impact on the understanding and application of such psychrophilic enzymes”: why?
Response 89: The statement was updated in the revised manuscript.
- Lines 355-358: “we anticipate a quick and efficient purification process such as microfluidic chip, mass spectrometry, NMR and integrative structural biology approaches will allow subsequent 3D structure elucidation to get along with other fields of studies.” This statement is general and not an anticipation at all, as this multidisciplinary approach is applied since more than 15 years in the field of characterization of molecular mechanism of protein properties.
Response 90: The sentence was rephrased and updated in the revised manuscript.

Round 2
Reviewer 1 Report
No further comments. The paper can be published as is.
Author Response
We thank the reviewer for the constructive comments on our manuscript. Indeed, those comments and suggestions have improved our manuscript.
Reviewer 2 Report
The language and contents have been improved. The contents are more coherent with the title. There are still language issues to be refined throughout the text (i. e., Line 480 pag. 14: "was unable to observe" instead of "was unable observed").
Conceptual things to fix:
- Fig. 1 did not improve that much. Images are not informative with respect to major drowbacks and, in principle, isolation and 3D structure determination cannot be considered "drowbacks" during enzyme production. At least the caption has to be modified
- Line 482 pag. 14: in the statement "The cold-active lipases followed the acylation and diacylation, forming a" "diacylation" is a wrong term, the correct one is "deacylation"
- In the paragraph entitled "Three-Dimensional (3D) Structure and Functional Mechanisms of Cold-active Lipase and Esterase" the amount of information given to the possible cold-adaptation mechanisms is still too scarce.
- Line 467-468, pag. 14. "Noby, Johnson [145] also attributed the substrate specificity to a plug at the end of an acyl binding pocket that blocks access to a buried water-filled cavity". How is this statement related to a cold-adaptation mechanism of the enzyme? It seems to be related with its substrate specificity
- Line 472 pag. 14: "Tolerance to changes in water entropy" Entropy increases or decreases? Is there any hypothesis why this should improve cold-activity of esterase at molecualr level? Please, provide a better explanation on this point
- Lines 473-474: "improved substrate accessibility to active site, since enzymes such as esterase with the active site located at the end of molecular tunnels, the substrate must first navigate the channel binds for upward catalysis" Why an increased accessibility of the active site is associated to a higher cold-activity? Not all esterases have "tunnels", but it is common they have hydrophobic pockets that are convered with flexible lids to avoid the exposure of hydrophobic residues to the solvent. This part is confusing and needs revision. Moreover, the authors must provide more examples to argue that cold-active esterases/lipase have these pockets usually higher exposed.
-Line 482 "The cold-active lipases followed the acylation and diacylation, forming a tetrahedral intermediate" This part must be merged with the first part of this paragraph, when the authors describe the general mechanism of catalysis of esterase, as this is the same of lipases.
- Line 484-488 "Despite pointing out the challenge in understanding the temperature adaptation of two similar enzymes based on their 3D structures, and further suggests a combined effect of the mutations might likely change the activation enthalpy and entropy like those seen in other cold-adapted enzymes, the only single mutation in the binding site cannot be ignored, since the binding pocket is also responsible for the cold adaptation as earlier stated" This statement is pointless and should be removed or completely rewritten, because it has bad construction and it is not clear what it is referred to.
- The authors could enrich the description of the molecular mechanisms for esterases/lipases cold-adaptation with a mention on recent works that demonstrated the involvement of metal ions in certain cases, both by increasing the cold-activity at the expenses higher active-site thermolability (https://doi.org/10.1038/s41598-020-61217-6), and by improving the stability of the active site at sub-optimal temperatures (https://doi.org/10.1111/febs.16661).
Past research indicated that there is not a directional trend in the cold-adaptation mechanisms of cold-active enzymes, including esterases. Many solutions arose during protein evolution, sometimes leading to opposite side-effects. Therefore, a statement should be added to underline this concept in ths study of cold-adaptation mechanisms of enzymes, also in the conclusion section
Author Response
We appreciate the time and effort and dedication that you put into our manuscript, you indeed provide valuable feedback and constructive suggestions.
Please, find below our responses to the reviewer’s comments:
The language and contents have been improved. The contents are more coherent with the title. There are still language issues to be refined throughout the text (i. e., Line 480 pag. 14: "was unable to observe" instead of "was unable observed").
Response 1:
Thank you for the valuable feedback that resulted in improving our manuscript.
We have checked the language issues and revised them accordingly throughout the manuscripts.
Conceptual things to fix:
- Fig. 1 did not improve that much. Images are not informative with respect to major drawbacks and, in principle, isolation and 3D structure determination cannot be considered "drawbacks" during enzyme production. At least the caption has to be modified
Response 2:
The caption has been modified to “Summary of various sections of this review” and updated in the revised manuscript.
- Line 482 pag. 14: in the statement "The cold-active lipases followed the acylation and diacylation, forming a" "diacylation" is a wrong term, the correct one is "deacylation"
Response 3:
Thank you for the correction. We have replaced the word ‘diacylation’ with ‘deacylation’.
- In the paragraph entitled "Three-Dimensional (3D) Structure and Functional Mechanisms of Cold-active Lipase and Esterase", the amount of information given to the possible cold-adaptation mechanisms is still too scarce.
Response 4:
Thank you for making the suggestion. Some information has been added to this paragraph and updated in the revised manuscript.
- Line 467-468, page. 14. "Noby, Johnson [145] also attributed the substrate specificity to a plug at the end of an acyl binding pocket that blocks access to a buried water-filled cavity". How is this statement related to a cold-adaptation mechanism of the enzyme? It seems to be related to its substrate specificity
Response 5:
Thank you for putting us back on track. The statement was indeed related to substrate specificity than the cold adaptation mechanism. We’ve removed the statement and updated it in the revised manuscript.
- Line 472 page. 14: "Tolerance to changes in water entropy" Entropy increases or decreases? Is there any hypothesis why this should improve cold-activity of esterase at molecular level? Please, provide a better explanation on this point
Response 6:
Thank you for pointing it out. More explanation has been included in the revised manuscript. The hypothesis of tolerance to the shift in water entropy of cold-active enzymes has been included with the following citations (Siddiqui and Cavicchioli, 2006, Kumar and Nussinov, 2004).
- Lines 473-474: "improved substrate accessibility to active site, since enzymes such as esterase with the active site located at the end of molecular tunnels, the substrate must first navigate the channel binds for upward catalysis" Why an increased accessibility of the active site is associated to a higher cold-activity? Not all esterases have "tunnels", but it is common they have hydrophobic pockets that are convered with flexible lids to avoid the exposure of hydrophobic residues to the solvent. This part is confusing and needs revision. Moreover, the authors must provide more examples to argue that cold-active esterases/lipase have these pockets usually higher exposed.
Response 7:
Thank you. The section was revised and updated in the revised manuscript.
-Line 482 "The cold-active lipases followed the acylation and diacylation, forming a tetrahedral intermediate" This part must be merged with the first part of this paragraph, when the authors describe the general mechanism of catalysis of esterase, as this is the same as lipases.
Response9:
The part has been removed and merged as suggested.
- Line 484-488 "Despite pointing out the challenge in understanding the temperature adaptation of two similar enzymes based on their 3D structures, and further suggests a combined effect of the mutations might likely change the activation enthalpy and entropy like those seen in other cold-adapted enzymes, the only single mutation in the binding site cannot be ignored, since the binding pocket is also responsible for the cold adaptation as earlier stated" This statement is pointless and should be removed or completely rewritten, because it has bad construction and it is not clear what it is referred to.
Response 10:
Thank you. The statement has been reconstructed and updated in the revised manuscript.
- The authors could enrich the description of the molecular mechanisms for esterases/lipases cold-adaptation with a mention on recent works that demonstrated the involvement of metal ions in certain cases, both by increasing the cold-activity at the expenses higher active-site thermolability (https://doi.org/10.1038/s41598-020-61217-6), and by improving the stability of the active site at sub-optimal temperatures (https://doi.org/10.1111/febs.16661).
Response 11:
Thank you for the suggestion. We’ve included the description in the paragraph of the revised manuscript.
Past research indicated that there is not a directional trend in the cold-adaptation mechanisms of cold-active enzymes, including esterases. Many solutions arose during protein evolution, sometimes leading to opposite side-effects. Therefore, a statement should be added to underline this concept in this study of cold-adaptation mechanisms of enzymes, also in the conclusion section.
Response 12:
The statement has been added under the cold-adaptation mechanism section as well as the conclusion section.
References:
KUMAR, S. & NUSSINOV, R. 2004. Different roles of electrostatics in heat and in cold: adaptation by citrate synthase. Chembiochem, 5, 280-90.
SIDDIQUI, K. S. & CAVICCHIOLI, R. 2006. Cold-adapted enzymes. Annu. Rev. Biochem., 75, 403-433.
Round 3
Reviewer 2 Report
The manuscript has been adequately improved to worth publication.
Some minor language checks are required, but conceptually issues have been fixed.
In the following a suggestion to improve the understanding of a section of the manuscript:
- Lines 342 to 358: "An increased surface negative charge is thought to play a role in addressing water entropy through the retention of stable hydrophobic interactions and solvation with water despite fluctuations in entropy and viscosity [137]. Adjustment to shift in water entropy of cold-active enzymes has been postulated earlier [138, 139]. The second mechanism of improved substrate accessibility to the active site involved the formation of additional tunnels to access the active site and increase the active-site cavity volume to a larger fold, which was absent in esterase from other closest homologues. Since some of the esterases with active sites located at the end of molecular tunnels, the substrate must first navigate the channel binds for upward catalysis. Increased cavity volume has been suggested as a mechanism for cold adaptation that can facilitate substrate binding at low temperatures [140, 141]." may be improved with
"An increased surface negative charge is thought to play a role in addressing water entropy through the retention of stable hydrophobic interactions by inscreasing the interactions of surface residues with water, despite fluctuations in entropy and viscosity [137]. Adjustment to shift in water entropy of cold-active enzymes has been postulated earlier [138, 139]. In esterases with active sites located at the end of molecular tunnels it was noticed that cold-activity is related to improved substrate accessibility to the active site by the formation of additional tunnels to access the active site and by increasing the volume of the active-site cavity. This was noticed by comparing cold-active esterases with other mesophylic or thermophylic closest homologues [140, 141]."
Author Response
We appreciate your comments. The section has been rewritten as suggested and updated in the revised manuscript. We appreciate your comments.